# Attitudes That Might Impact upon Donation after Brain Death in Intensive Care Unit Settings: A Systematic Review

**DOI:** 10.3390/healthcare11131857

**Published:** 2023-06-26

**Authors:** Areej Alshammari, Michael Brown

**Affiliations:** 1Northern Area Armed Forces Hospital, Hafar Albatin, King Khaild Military City 39748, Saudi Arabia; 2School of Nursing and Midwifery, Queen’s University Belfast, 97 Lisburn Road, Belfast BT9 7BL, UK

**Keywords:** organ donation, organ donor, tissue and organ procurement, brain death, organ harvest (sic), brain dead, critical care, intensive care

## Abstract

Background: Organ donation and transplantation can save or improve the quality of life of people worldwide. However, there are many challenges associated with organ donation, as the demand for organs greatly outstrips supply. Additionally, there are brain-dead patients who could be potential organ donors. It is thus important to determine the attitudes affecting organ donation and transplantation in intensive care unit settings. Aim: this study aimed to identify attitudes that might affect organ donation and transplantation in intensive care unit settings. Methodology: Five electronic databases (CINAHL, Medline, PsycINFO, Scopus, and EMBASE) were searched systematically. A systematic search strategy was formulated. The quality of each study was assessed using the MMAT quality appraisal tool. Results: A total of seven studies were included. The findings of this systematic review demonstrate that education, policy, and continuing professional development could help to address barriers to donation. Conclusions: therefore, to influence organ donation and transplantation positively, the main themes evaluated in this systematic review provide an opportunity to influence organ donation and transplantation attitudes in intensive care unit settings.

## 1. Introduction

Maintaining or improving quality of life is a key driver in healthcare innovation. Due to this, there is continuous research and development within the rapidly evolving fields of science and technology, with research evidence highlighting the latest treatments and methods for both improving and saving human life. One key threat to quality of life, or life itself, is organ failure [1,2]. Organ donation and transplantation can save multiple lives, involving taking healthy organs and tissues from one person, gaining legal consent for the procedure, and transferring them to several other individuals for transplantation [3]. Research evidence shows that one organ donor can save or significantly improve the quality of up to ten lives [4,5]. Furthermore, significant scientific and technological advances regarding the benefits of organ donation and transplantation have been achieved in the field of organ donation and transplantation [1]. Despite this, organ donation and transplantation continue to experience significant challenges. The demand for transplantation substantially outstrips supply, and waiting lists are expanding every single day, as asserted by researchers [1,6,7]. The issue has become a global concern due to reports that an increasing number of patients die annually waiting for an organ transplant [8,9,10]. In the United Kingdom, around 20% of patients who were on the transplant waiting list in 2017 were removed due to either their condition deteriorating past the point of the procedure or due to death [11]. In addition to this, based on the United Network for Organ Sharing (UNOS), deceased donation increased by 39% in the last five years; however, the need for organs is still immense, with a total of 104,047 people waiting for a lifesaving organ transplant [12]. Wakefield et al. reported that where a requisite number of organs are available, a significant number of patients worldwide stand to benefit from transplant procedures [5]. However, many patients do not receive the organ that they need, as the number of organ donors is low even in countries with a high success rate in transplantations due to a variety of factors, including certain restrictions that are applied to waiting lists. Fleming and Thomson [13] attribute the variations in organ donation rates between nations to a range of factors, including the religious, social, legal, cultural, and organisational contexts of the specific country. Fleming and Thomson also emphasise the critical role played by doctors in improving the number of organ donors [13]. To be eligible for organ donation, the patient must be determined to be dead with regard to biological or neurological criteria; therefore, critical care management is vital, and many national regulations have been updated to help reduce the stark variations in organ donation and transplantations between nations [14]. Organ donation is a lifeline for patients who are in the end-stage of organ failure. However, the ongoing situation worldwide is that many potential and suitable donors have not provided advanced consent for the donation of their organs. This means that when patients are declared brain dead, either the donation possibility is lost, or the family is left with a difficult decision at an emotional time. Therefore, it is important to determine at an early stage the attitudes towards and level of knowledge of organ donation and transplantation to identify the factors that may affect the likelihood of donation. This review, therefore, aims to identify the attitudes that might impact organ donation and transplantation in intensive care unit settings and make evidence-based recommendations that may increase organ donation, especially in countries with low organ donation rates.

### Aim

The aim of this review was to identify the attitudes that might impact organ donation and transplantation in intensive care unit settings and make evidence-based recommendations that may increase organ donation.

To address this aim, three questions were designed to guide the review: (i) What are the facilitators that enable organ donation for brain-dead patients in intensive care unit settings? (ii) What are the potential barriers to organ donation for brain-dead patients in intensive care unit settings? (iii) What are the available strategies to increase the number of potential organ donations from brain-dead patients in intensive care unit settings?

## 2. Materials and Methods

### 2.1. Search Strategy

The Preferred Reporting Items for Systematic Reviews and Meta Analysis (PRISMA) checklist was used to develop the quality and transparency of the data included and to guide the reporting of the systematic reviews [15,16]. The systematic search study was formulated with the assistance of a subject expert librarian. A brief scoping review of research on organ donation and transplantation was conducted prior to the preparation of the search strategy to determine the terminology common within these topics. 

Electronic searches were conducted of five major research databases. The databases searched were CINAHL, Medline, EMBASE, PsycINFO, and Scopus. The time frame for the review was 5 years to reflect the changes and developments in intensive care practice regarding organ donation. The following search terms were used: ‘organ donation’, ‘organ donor’, ‘tissue and organ procurement’, ‘organ harvest (sic)’, ‘brain death’, ‘brain dead’, ‘critical care’, and ‘intensive care’. Terms were searched in the English language only. As well as searching the databases individually, a combined search was also consequently completed. Endnote version 20 was used to manage electronic references and to streamline the process for removing duplicates. Furthermore, in order to broaden the search, the Boolean ‘OR’ feature was added. The Boolean operator ‘AND’ was used to help in narrowing the search to identify any relevant articles in each database. 

### 2.2. Eligibility Criteria

Articles were included if they met the following criteria: Published between the years 2017 and 2022.Written in the English language.Quantitative research design.Focusing exclusively on brain death.Focusing exclusively on intensive care unit setting.Focusing exclusively on attitudes.Reporting on adult medicine.

Papers were excluded if they met the following criteria: Published before the year 2017.Not written in the English language.Engaging with cardiac death rather than brain death.Not based in intensive care unit setting.Reporting on paediatric medicine.Used qualitative or mixed-method study design.Other type of literature including review, conference proceedings, abstract, dissertation, editorial, or researcher commentary.

### 2.3. Selection of Studies

The systematic search of the five databases resulted in a total of 1043 studies, with 267 duplicates removed. The abstracts of the remaining studies were reviewed against the inclusion and exclusion criteria. Of these, 410 studies were removed as they did not meet the criteria and the aim of the review. Finally, the full texts of the remaining two hundred and six papers were read and assessed against the aim and review criteria. Seven studies met both the inclusion criteria and the aim of the review. A research PRISMA flow chart detailing the search is presented in Figure 1.

### 2.4. Quality Assessment Tool

A quality assessment was conducted using the Mixed-Methods Appraisal Tool (MMAT) to evaluate the methodological strength of the studies. The MMAT was chosen as the critical appraisal tool for this review. The MMAT contained five categories regarding the study designs, with each criterion rated as ‘yes’, ‘no’, or ‘cannot tell’ [17]. 

### 2.5. Data Analysis, Synthesis, and Presentation 

Due to the homogenous nature of the studies, a descriptive narrative synthesis of the findings of each study was performed. The initial synthesis involved appraising and searching the studies, and listing and presenting the findings in tables. After that, the findings were categorised into three thematic categories based on their common characteristics of studies. Subsequently, the included studies were summarised in a narrative synthesis, which was agreed upon by the reviewers, and the differences were discussed and resolved.

## 3. Results

### 3.1. Characteristics of Included Studies

Seven studies were included in the review. Two studies were conducted in Turkey [18,19], three were conducted in South Korea [20,21,22], one was conducted in Poland [23], and the final study was conducted in France [24]. Two studies collected data across multiple hospitals [23,24], and the other studies were located in a single site [18,19,20,21,22]. Some studies focused on investigating the experiences and attitudes of different populations of healthcare providers towards organ donation and transplantation: healthcare providers [23,24] and relatives of the patient [18,20]. In three studies, the data were obtained from the medical records of patients who were diagnosed with brain death [18,19,22]. All of the included studies were conducted by professional researchers and research teams from a range of academic backgrounds. The studies showed variations in sample size, from 54 to 3325 participants. The smallest sample size was *n* = 54, found in the study by [22]. The largest sample size was found in the cohort study conducted by Kentish-Barnes et al., with a sample of n = 3325 participants [24]. Additionally, there were variations in sample size, dependent on the participant population. However, it was noted that all of the studies were from countries with low deceased donation rates, except the study from France. Along with this, for the studies examining the attitudes and potential barriers to organ donation from the perspective of patients’ relatives, the number of participants ranged from 92 to 102 [18,20,21]. The samples ranged from 54 to 3325 for those studies focusing on healthcare professionals’ views and attitudes towards organ donation and transplantation [23,24] (see Table 1).

All seven studies included used quantitative methods. In three studies, the data were collected using survey methods [20,21,23]. One study used a cross-sectional design with no control group, with the researchers using factorial correspondence analysis [24]. Three studies collected the data retrospectively [18,19,22].

### 3.2. Quality Assessment of the Studies Included

Four studies were rated as being ‘strong’ [18,19,22,24] and three were rated as being moderate [20,21,23] using the MMAT quality appraisal tool. The review sets out the quality of the individual studies. All seven of the studies had similar limitations and strengths, such as limited sample size in three studies [20,21,22] and strength in another [19]. Questionnaire validity was a limitation of the study conducted by Lee et al. [20]. While it is essential for all researchers to adopt ethical practices, bias can affect studies in many ways. For example, the generalisability of results based on convenience and purposeful sampling is considered to be a key limitation of a study due to an increased risk of bias [25]. Therefore, several studies used different methods to enhance their potential impact and increase reliability and credibility. The following section analyses each of the included studies in turn, assessing the individual strengths and weaknesses (see Table 2).

Based on the analysis of the studies, three main themes were identified: (i) education implications, (ii) policy implications, and (iii) continuing professional development implications. The most recurring theme was focused on educational implications, followed by policy implications and continuing professional development (CPD) implications. Most of the study findings related to more than one theme.

#### 3.2.1. Theme 1: Educational Implications

The studies included in this review indicate that education is necessary to increase the organ donation rate. Both public and institutional education programmes need to equip the current and future generations of health professionals and the wider community with a clear awareness and understanding of the essentials regarding organ donation. 

##### Public Education

Two studies focused on the importance of educating the public, notably, the family members of future potential donors. This is important as many patients and their family refuse donations due to a lack of knowledge. Park et al. investigated how phased education can influence family members’ attitudes towards organ donation [21]. These findings support the need for public education programmes that target all parts of society, with family counselling and support being routine parts of the donation process. They emphasise the importance of providing clear information to family members from the first visit to the ICU regarding the patient’s condition, ensuring that they clearly understand the concept of brain death. Moreover, any education intervention must also include the importance of sharing decisions in advance about donations with family members, to help to improve the organ donation rate. Additionally, Arslantas and Çevik highlighted that increasing awareness through education is necessary in order to increase consent to donate [19]. 

##### Undergraduate Education

Two studies discussed the importance of including organ donation when preparing the next generation of healthcare professionals during their education programme. It has been identified as being important by Kentish-Barnes et al. and by Szydlo et al., who note that organ donation should be included within medical education to establish a firm knowledge baseline [23,24]. It is anticipated that education will indirectly improve the organ donation rate. 

#### 3.2.2. Theme 2: Policy Implications

Five studies reported the need for clear policies regarding organ donation to help the healthcare team to manage the donation process. They concluded that evidence-based strategies around perceptual barriers and protocols for optimising organ donation are key aspects of organ donation policies. Establishing clear strategies and policies to identify barriers based on negative perceptions of organ donation would increase the rate of donation effectively [26]. 

##### Protocols to Optimise Organ Donation Process

The need for coherent policies and protocols was highlighted in studies by Tore Altun et al. and Park et al. regarding the management of brain-dead patients and potential organ donors [18,22]. Park et al. emphasised the early management of fluid challenges, adequate usage of a vasopressor when monitoring cardiac output, early CRRT, and modified apnea tests in increasing the conversion rate of potential donors to actual organ donors [22]. The implementation of clear policies and protocols will reduce the number of organ mismatches and thus failed outcomes [22]. Furthermore, Arslantas and Çevik reported that having policies in place concerning the obtaining of consent are essential in increasing organ donations [19]. 

#### 3.2.3. Theme 3: CPD Implications

Healthcare professionals play a key role in the organ donation process [27,28,29]. Knowledge and experience of the process is required to support the increase in demand for organ donation [30]. This review identified several CPD implications: organising conferences and workshops and holding regular meetings. 

##### Information Sharing

Four studies stressed the importance of organising conferences and workshops on the topic of organ donation for healthcare professionals. Szydlo et al. assert that anaesthesiology and intensive care doctors need dedicated workshops to educate them on how to determine brain death and care for potential donors [23]. This would also improve clinical skills and experience and increase confidence regarding organ donation. Tore Altun et al. and Kentish-Barnes et al. underscore the need for donation-specific education programmes and workshops for healthcare professionals [18,24]. Park et al. emphasise the need for workshops on managing brain-dead patients, focusing on fluid challenge, the adequate usage of vasopressor under the monitoring of cardiac output, early CRRT, and modified apnoea test [22], as well as having dedicated intensive care doctors, who are on hand to manage patients and increase the rate of organ donors. Kentish-Barnes et al. emphasised the positive effect of holding regular clinical meetings [24]. By having support groups and a positive work environment, healthcare professionals are afforded the chance to discuss any future improvement required to develop the organ donation process. 

## 4. Discussion

Organ donation and transplantation is considered the optimal treatment for many patients who are diagnosed with end-stage organ failure [8,31,32,33]. Organ transplants are performed worldwide with a total of 144,302 solid organ transplants carried out every year according to 2021 data from the Global Observatory on Donation and Transplantation. Also, the organ donation operations completed in 2021 were 8409 heart transplants, 6470 lung transplants, 92,532 kidney transplants, 2025 pancreas transplants, and 34,694 liver transplants [34]. Despite these impressive figures, many patients still do not receive the organs that they need, waiting lists are lengthened annually, and patients die during their elongated wait [35,36,37,38]. There are many various causes of this global shortage of organs. While significant advances in medicine over the past twenty years have made organ donation and transplantation procedures much more successful and part of clinical routine, there remains a severe problem related to access to the organs needed. It is this issue that the author is interested in addressing. 

### 4.1. Educational Implications

Researchers have indicated that the decreased rate of organ donation is due to the lack of awareness and knowledge among various professionals, for example, health workers, nurses, teachers, and students [1,7,39,40]. Şenyuva strongly indicated that a lack of knowledge and awareness is one of the factors that causes a shortage in organ donation, and it must be taken seriously [1]. The finding shines a light on the need for education in order to increase the organ donation and transplantation rates. Şenyuva also asserted that most of the study participants wanted to be organ donors, yet did not know the processes involved or how to apply [1]. Therefore, utilising visual media and the Internet become essential in increasing knowledge regarding organ donation [1]. The importance of educating the public to increase organ donation is vital, as is the need to ensure that undergraduate students have an awareness and understanding of the issues regarding organ donation [1]. To support these efforts, it has been identified that educating students about organ donation at schools and colleges can have a positive effect on their views regarding organ donation [41,42,43,44,45]. Therefore, Timar et al. suggest that education be provided by presenting information to student groups and classes through special programs that encourage students to discuss organ donation and the process [46]. Indeed, Venkatesan et al. strongly asserted that around 74% of the teachers who participated in the study were of the view that organ donation must be added to the school curriculum [47]. Hence, Tsubaki et al. reported, based on the review that he conducted, that undergraduate nursing students worldwide are accepting organ donation more compared with others, and this was supported by a study in the United Kingdom as well, which reported that nursing students are more likely to hold an organ donor card [44].

Ormrod et al. found that some family members refused to provide consent for organ donation as they did not receive appropriate and adequate education and information [48]. Along with this, Zhang et al. and SPITE strongly indicated that proper education is essential, as studies regarding the attitude towards organ donation have suggested that the refusal was caused due to incorrect information and unsatisfactory knowledge [49,50]. In other words, as Tontus states, there are strong relationships between accepting, the knowledge of organ donation, the willingness to donate organs, and the level of awareness [51]. Education is also required by those registered as organ donors about the importance of informing their family about their end of life wishes [52]. Morgan et al. reported that during discussions about organ donation, some participants were surprised to learn that family permission was required for donation to proceed, with the wishes of the potential organ donor being ignored [53]. 

### 4.2. Policy Implications

In recent years, policymakers have made repeated attempts to address the organ donation shortage through adopting policies designed to improve the administrative efficiency of the organ donation system and process [54,55,56]. However, identifying the precise barriers which affect organ donation and updating policies and practices to address these remain challenges [46,57]. In order to identify the maximum number of organ donors, transplant coordination networks need to be expanded to be more effective and functional. A good example of this in action would be the expansions which have taken place in Poland since 2010 [58]. The researchers underscore the importance of including every hospital to improve the quality of organ donation by identifying and monitoring potential organ donors and evaluating how effective current policies are increasing donations [58].

Policies regarding the role of transplant coordinators are required to streamline the information process. The role should involve reports and updates for hospital managers, including the assessment rates for the potential donations, incorporating the number of deaths, the number of deaths with suspected brain death, the number of confirmed brain deaths, the number of meetings with family members conducted, and the number of actual organ donors. Such data would help policymakers to identify areas for improvement and development, as they seek to update existing policies [58,59]. Additionally, there is a need to address the needs of family members through developing clear communication policies and strategies, as indicated by [60,61,62]. Societal taboos around death are one barrier to discussing organ donation with family members [63]. Fear over discussing death is a fundamental barrier to organ donation, which researchers identify as being particularly intractable [53,64]. Moreover, policymakers need a comprehensive understanding of the role that religion plays in attitudes towards organ donation, and how this could be a barrier to potential donations [53]. 

### 4.3. CPD Implications

Healthcare professionals are key personnel in the organ donation process, since their role is critical in increasing the number of organ donors [1]. Akbulut et al. agreed that doctors are vital in identifying potential donors, communicating with organ donor co-ordinators, and gaining consent from family members [65]. Moreover, other health professionals, including registered nurses, also provide information and support to family members about the organ donation process, answer their questions, and help to identify potential organ donors [1]. And, based on this, CPD training for healthcare professionals is important to help to improve the understanding of the concept of brain death, the organ donation process, and how to manage the care of potential organ donors without forgetting about the need for sensitive communication from all healthcare professionals in supporting family members. Continuous training leading to more successful organ and tissue donations was strongly highlighted by [66,67]. Therefore, continuous education and professional development are required for all healthcare professionals to ensure that their contributions are maximised [46]. Conferences, expert panels, seminars, and other educational events need to be organised on the topic of organ donation [1]. In one study, Oczkowski et al. found that doctors have limited communication skills in influencing a family’s decision to provide consent [68]. Therefore, specific training for doctors in intensive care units is a necessity [46]. Organ transplant coordinators play a critical role during interactions with family members during the decision-making process. A well-trained co-ordinator can support family decision-making regarding organ donation [69]. However, Darnell et al. argued that many families refused to donate their family member’s organs if the coordinator failed to recognise the family’s needs, educational background, or religious background [70]. This finding was strongly supported by Jacoby et al., who highlighted the crucial role that coordinators play in family decision-making [69]. Therefore, CPD training needs to be targeted and concentrated to achieve the goal of increasing the number of organ donations, while improving the experience of family members during this emotional time [70]. 

### 4.4. Implications for Future Research

It is important that further research determines whether an increase in positive attitudes towards organ donation in the community increases the pool of potential organ donors available. Potential donors are another subject of study for future research. Interviews and qualitative methods should be used to examine the education process of becoming a donor and to explore the factors that motivate individuals to register to donate. Moreover, more research needs to be undertaken to investigate whether academic institutions have introduced the organ donation process into the curriculum and how far this has shaped undergraduate students’ knowledge and willingness to donate. More broadly, the continuous evaluation of the effectiveness of the policies and procedures in place across a variety of organisations would be helpful in monitoring potential organ donors, the effectiveness of current communications, and the roles of organ donation coordinators and teams. Multi-centre studies should be conducted to provide standardised guidelines for the management of brain-dead patients to help in increasing the rate of organ donation. Future research must consider continuous monitoring of CPD training in order to improve it to both motivate the healthcare professionals involved in the organ donation process and to empower them to achieve positive outcomes.

### 4.5. Strengths and Limitations of this Review

One of the strengths of this systematic review is that it presents the most recent research and focuses on a quantitative research design, with all of the studies meeting the inclusion criteria included. However, this review also has several limitations. Five of the studies were conducted as single-centre studies, which may limit the generalisability of the findings. The sample sizes were small, posing a further challenge to the generalisability of the findings, with the questionnaires used in the studies not always being scientifically validated prior to use. The retrospective nature of some studies prevented them from determining any detailed reasons for not donating organs, for example regarding the reasons why family members did not consent to organ donation. Beyond this, the heterogeneity of the included studies made it difficult to compare findings and conduct a meta-analysis, and therefore a narrative analysis was undertaken. During the literature search, efforts were taken to be as comprehensive as possible in the use of terminology and rigour of the process. However, despite this, new terminology continues to emerge relating to organ donation and transplantation, and, therefore, related studies using different definitions may have been inadvertently overlooked. 

## 5. Conclusions

This systematic review focused on exploring attitudes that might impact organ donation and transplantation in intensive care unit settings. Organ donation and transplantation are recognised as the optimal treatment and cure for many patients around the world. The findings of this systematic review demonstrate that education, policy, and continuing professional development could help to address barriers to donation. Therefore, to influence organ donation and transplantation positively, administrators, organisations, educators, and policymakers must work together to achieve the goal of improving donation in low-donation-rate countries. Along with this, there is a need for proper education and the effective continuous development of it for all health care providers to keep them updated on the early identification and management of organ donors, without forgetting the critical role of organ donation co-ordinators. Overall, these were the main themes evaluated in this systematic review, as they directly influence organ donation and transplantation attitudes in intensive care unit settings. 

### Recommendations

This systematic review has highlighted that the rate of organ donation remains well below the required level to meet the global demand. To address these concerns, the following recommendations are given: Organ donation education needs to be continuous, well-planned, and implemented systematically.The public should receive targeted information, at all levels of education, to help them to overcome misinformation and positively influence attitudes towards organ donation.Organ donation co-ordinators should have access to intensive care education and continuous professional development activities to increase the impact of their role in the organ donation process.Conferences, panels, seminars, and educational events focused on organ donation and transplantations should be provided for healthcare professionals.Policymakers should keep reviewing and updating their policies and procedures regularly based on organ donation reports to find gaps and improve the rate of organ donation.

## Figures and Tables

**Figure 1 healthcare-11-01857-f001:**
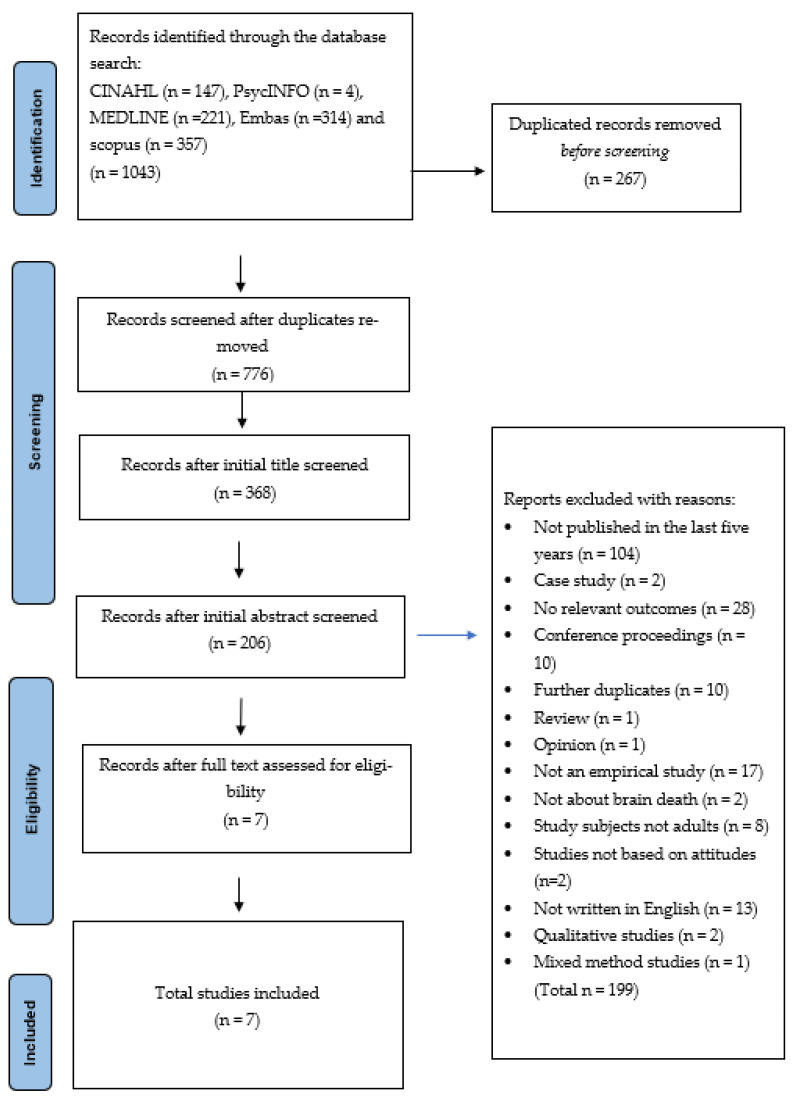
Research flow chart diagram.

**Table 1 healthcare-11-01857-t001:** Papers included in the review (n = 9).

Study Citation and Country	Aims	Sample	Methods	Key Findings	Recommendations	MMAT Score
[20] South Korea	To evaluate the attitude towards organ donation among relatives of patients in South Korea.	92 relatives of brain-dead patients in SICU.	Structured questionnaire (quantitative study design).	- Positive attitude regarding donating their own body (60.9%). - Positive attitude towards the medical staff recommendations (45.7%), while 15.2% felt it was disrespectful.	Evidence-based strategies need to be established. More research based on the general opinion of populations is required.	MODERATE
[21] South Korea	To investigate the effect of phase education in improving attitudes towards organ donation and willingness to donate.	92 family members of brain-dead patients in SICU.	Survey (quantitative study design).	The survey was performed in three phases and the results increased positively: - (Q1) (92.4%). - (Q2) (80.4%). - (Q3) (56.5%).	Provide public education and family counselling routinely.	MODERATE
[23] Poland	To identify factors and challenges influencing donation rates in intensive care units of various reference levels in region.	133 physicians caring for brain-dead patients in ICU.	Survey (quantitative study design).	- Agreeing with the brain death criteria (95.49%). - Lack of knowledge in declaring brain death (50%). - Need for workshops (94.76%).	Education and training to build up the knowledge and experience. Regular meetings and workshops can help to improve the organ donation rate.	MODERATE
[18] Turkey	To analyse the reasons why some potential donors did not become an actual organ donor, even with consent from the family.	102 potential organ donors.	Retrospectively (quantitative study design).	The actual transplantation rates for the organs were as follows: - Kidneys (88%), lungs (13%), heart (30%), and liver (70%). Reasons why organs were not transplanted: - Donor unsuitable (24%). - Logistical problems.	Develop protocols in optimising organ donation and reduce the mismatch between organs.	STRONG
[19] Turkey	To analyse potential brain death and factors associated with organ donation among patients.	Medical records of 629 patients in ICU. 102 brain-dead patients in ICU.	Retrospectively (quantitative study design).	- Patients died before performing the diagnostic test (18%). - Reasons for non-organ retrieval: Family refusal (89.5%). No relatives present (7%). Medical instability (3.5%).	To establish a strategy to increase consent and awareness as well regarding brain death and organ donation.	STRONG
[22] South Korea	To report their experience of brain-dead patient management by a dedicated intensivist who had wide experience in the treatment of hemodynamically unstable patients and to suggest the role of an intensivist in organ donation.	54 actual organ donors.	Retrospectively (quantitative study design).	- The mean time between admission until diagnosing the potential brain death: 6.28 ± 3.63 days. - The mean duration of brain death management after transferring the patients to the management team was 2.81 ± 1.21 days. - The mean number for recovered organs in each donor was 3.98 ± 1.55.	National support is needed as well as management protocols for brain-dead patients.	STRONG
[24] France	To describe ICU clinician’s perceptions and experience of organ donation.	3325 physicians and nurses working in ICU.	Cross-sectional study (quantitative study design).	- Professionals who described their experience as motivating were younger. Most of them were organ donors and participating in the meetings of transplant coordinators with relatives.	Improve the environmental culture, training support groups, and positive work environment.	STRONG

**Table 2 healthcare-11-01857-t002:** Results of the MMAT process on the included studies.

Number	First Author	Citation	Screening		Category	Q 1	Q 2	Q 3	Q 4	Q 5	Comments
			Q1	Q2							
1	Hyun Ji Lee	[20]	Y	Y	Quantitative descriptive study	Y	N	N	Y	Y	MODERATE
2	Ui Jun Park	[21]	Y	Y	Quantitative descriptive study	Y	N	CT	Y	Y	MODERATE
3	Marcin Szydio	[23]	Y	Y	Quantitative descriptive study	Y	Y	CT	CT	Y	MODERATE
4	Gulbin Tore Altun	[18]	Y	Y	Non-randomised study	Y	Y	Y	Y	CT	STRONG
5	Reyhan Arslantas	[19]	Y	Y	Non-randomised study	Y	Y	Y	Y	CT	STRONG
6	Jin Park	[22]	Y	Y	Non-randomised study	Y	Y	Y	Y	CT	STRONG
7	Nancy Kentish-Barnes	[24]	Y	Y	Non-randomised study	Y	Y	Y	Y	CT	STRONG

Yes = Y, no = N, cannot tell = CT.

## Data Availability

Not applicable.

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
