# Peer review of "Attitudes That Might Impact upon Donation after Brain Death in Intensive Care Unit Settings: A Systematic Review"

_healthcare, 2023, doi:10.3390/healthcare11131857_

Round 1
Reviewer 1 Report (Previous Reviewer 1)
Review 2 of this paper
Attitudes that might impact upon donation after brain death in intensive care unit settings: a systematic review.
Major Comments
This is an improved paper in its scope – thank you for engaging with the changes suggested in the review process.
Needs some areas addressed still.
Please have a read over all your new sentences (in red on my version). Multiple grammar mistakes and typos are present. I have indicated some as I found them below, but more may be present.
Abstract
‘However, this will be supported only with clear policies on improving the administrative efficiency of the organ donation system and process.’
Conclusion
‘However, this will be supported only with clear policies on improving the administrative efficiency of the organ donation system and process from identifying the potential organ donor to a successful transplantation.
This really what you want to say? Efficiency seems very harsh when your aim is attitude change. Some terrible ethical failings and crimes in history have been very efficient.
I much prefer your first submission wording but have modified to make fit the attitude focus now.
‘Therefore, to influence organ donation and transplantation positively the main themes evaluated in this systematic review provide an [opportunity/avenue] to influence organ donation and transplantation attitude in intensive care unit settings.’
- feel free to make this sentence your own. But it is positive and is a good summary of what you did and why.
Recommendations
I much preferred your old ones.
You raise new ideas right at the end not in keeping with your review and not justifiable by the logic of your own paper.
Where did 'correcting information’ a very authoritarian statement appear from?
As well as ‘hotline’ no study you reviewed had this.
‘Mandatory course containing the needed information about being an organ donor must be before issuing the Card donor, it would be helpful.’ Not sure what you mean but are you saying a member of the public needs to do a mandatory course before they can get an organ donor card? Wow. And no reference in your review did such a thing.
Suggest from your old recommendations (and your new number 3 which I liked) say:
1. Organ donation education needs to be continuous, well-planned and implemented systematically.
2. The public should receive targeted information, at all levels of education, to help overcome misinformation and positively influence attitudes towards organ donation.
3. Organ donation co-ordinators should have access to intensive care education and continuous professional development activities to increase the impact of their role in the organ donation process.
4. Conferences, panels, seminars and educational events focused on organ donation and transplantations should be provided for health care professionals.
5. Policy makers should keep reviewing and updating their policy and procedures regularly based on the organ donation reports to find the gaps and improve the rate of organ donation.
Minor Comments
Abstract.
Grammar needs correcting.
‘Aim: This study aimed to identify attitudes might affecting organ donation’
‘without forgetting the importance of critical role of the organ coordinators.’
This is a harsh and very utilitarian conclusion ‘However, this will be supported only with clear policies on improving the administrative efficiency of the organ donation system and process.’ That really what you want to say?
Keywords
You still have ‘organ harvest’ in your key words. This term is offensive. I appreciate it is in your search terms but you should add (sic) to indicate you are aware of its offensive nature. But there is no place anywhere else for it to appear.
In my correspondence to the editor I will be making this point and suggesting the journal should not be publishing papers which include offensive terminology – it falls into the same category in my opinion as using a racist term. Its only role is historical in the same way one might need to research racist language.
Introduction
Grammar
‘In addition to that and based on the 47 United Network for Organ Sharing (UNOS), the deceased donation increased 39% in the 48 last five years, however still the need for organs is immense with total of 104,047 people 49 are waiting for a lifesaving organ transplant [12]’
Methods
Do you mean intensive care ‘Focusing exclusively care unit settings.’
Results
‘seven’ needs capital for first word in results.
Conclusion
Grammar
“Along with that, there is a need for proper education and an effective continuous development for all health care providers and keep them updated on early identifying and managing the organ doners without forgetting the importance of critical role of the organ coordinators.”
- note also donor spells wrong as has an e
As above. Some improvement needed in the new sentences added between reviews.
Author Response
Thank you so much, all your feedback and suggestions was really helpful. please find in the attachment my replay to the report.

Reviewer 2 Report (Previous Reviewer 2)
Dear Authors,
I think that your work is improved after revision.
Changed title and abstract could attract potential readers. I think that you postulated a clearer aim for the work you have done focusing on attitudes toward organ donation and skipping the transplantation part.
The methodology and results sections are clear and easily understandable as they were before.
I congratulate changes you have made in the recommendation section.
Even though there is very little novelty in the results and recommendations, but as you mentioned, even the newest studies did not reveal anything unexpected, which is an important statement for the low-donation countries.
Please, fix typing mistakes throughout the corrections in the added text in red, e.g. Conclusions section where "organ coordinators" or "doners" are mentioned.
Author Response
Thank you so much, I appreciate your feedback. please find in the attachment my replay to the report.

This manuscript is a resubmission of an earlier submission. The following is a list of the peer review reports and author responses from that submission.
Round 1
Reviewer 1 Report
Factors that impact upon organ donation and transplantation in intensive care unit settings: a systematic review
Healthcare
Major Comments
This is a very confusing paper.
The English and grammar are fine, and it is an easy read.
It has however three big problems.
1. The title is not true.
The studies you identify do not link rates of organ donation to the factors you identify. At best they support a hypothesis. The conclusion of the French study says as much, “Significant differences exist among ICU clinician's perceptions of organ donation. Whether these differences affect family experience and consent rates deserves investigation.”
A truer title might concern attitudes.
“Attitudes that might impact upon donation after brain death in intensive care unit settings: a systematic review.”
2. The methodology is flawed.
You must have realised something was wrong with your methodology when only one country with a highly developed organ donation and transplantation program (France) was included in your review.
You searched for English papers but found no studies from an English-speaking country.
The 9 papers you identify are completely different. It’s all apples and oranges. Most were surveys, and who was surveyed varied considerably making strong conclusions very doubtful. Those that concerned process seemed forgotten in your conclusions and findings.
My limiting yourself to brain dead patients you miss big studies from countries that practice both DBD and DCD. For example: https://pubmed.ncbi.nlm.nih.gov/33860929/
And what about https://pubmed.ncbi.nlm.nih.gov/32693247/
Perhaps you are interested only in attitudinal factors and not factors with clear associations to rates of donation?
In which case you must not make sweeping generalisation and conclusions?
Especially as survey data – quantitative, lacks the nuance of all the good qualitative research that has occurred researching family attitudes. It may well be that attitude research is best carried out qualitatively than quantitatively. So your included studies are very narrow for the topic.
Fundamentally, your article selection is too broad (ICU staff attitudes, patient families on ICU who are responding theoretically, medical records). It makes coherent conclusions very difficult. Firstly you need to bring the similar research together in your paper so one can logically interpret them as a reader. Bring like with like together. Then you need to be much less dogmatic and conclusive.
3. Who is your audience?
This is not a very generalisable study. Who is your audience, to whom are you addressing your findings to? Small and medium size donating countries? Large countries? Those who educate the public? Those who educate ICU staff? Without a clear audience goal your paper has no use and lacks clarity of intention.
Is there a useful paper lurking within this paper?
Perhaps.
I would need to see:
A) A title that matches the papers found.
B) Either a narrowing of focus (removing the process papers and just focussing this paper on attitudes). Perhaps the easiest option for you since your conclusions are mostly all attitude based.
Or a complete rewrite that brings the like with like papers together and has conclusions on just that section, before moving onto the other papers. ie Discuss the process papers than the attitude papers and make separate conclusions.
C) A consideration of opening your search criteria so you don’t miss process papers like I have highlighted above (if you still wish to include process factors like consent, ethnicity, age and how they alter donation rates).
D) A clear audience in mind you are writing the paper for.
Minor Comments
CPD needs to be defined in abstract. But CPD, in broad terms, is just education for healthcare professionals. I don’t understand why you make it separate to education theme.
You use the offensive term ‘harvest’. I appreciate you only do so when describing search terms – but some acknowledgment of its offensive nature needs to be made. Please remove from key words. When describing the search term please use ‘Organ Harvest (sic)’
Author Response
thank you so much for your feedback, please see the attachment.

Reviewer 2 Report
The authors have chosen an important for better organ donation results topic for their systematic review of the literature and the intention to analyze only the most recent data is commendable.
Nevertheless, this review is regarding solely organ donation, and transplantation is not discussed in any part of the article. So, I would suggest abandoning connections to organ transplantation starting from the topic of the article and throughout the paper.
Abstract: the result part mentioned exclusively the main focus of the identified studies but not their findings, which should be a result of the analysis. This is what I would suggest adding. I would also suggest being more specific on conclusions. That would make the abstract attractive and understandable for the first-time reader.
The introduction and methodology are correct and I have no remarks.
Obtained results are presented clearly and visualization is adequate.
I am missing the author's view in the discussion and interpretation part on what has changed compared to previous studies because most of the results and interpretations are very similar to what we already know. Possibly the best example is the Spanish donation model speaking a lot about 360 education and the family approach (Matesanz et al. Transpl Int 2011).
Interestingly that all the studies are from very low deceased donation rate countries, except for 1 from France and relatively from Poland (www.irodat.org) maybe it is a reason why the data is repeated. Of course, some things always remain on the top of importance but still, this point of view could make this paper more interesting. Maybe authors could have a look at that?
I am also missing discussion and conclusions on donor management protocols even though one of the studies pointed out this topic. It is a primary goal in ICU. The authors could interpret on this more and probably correct conclusions.
Author Response

(The authors gave the same response as above.)
